# Simulation Research on the Relationship between Selected Inconsistency Indices Used in AHP

**DOI:** 10.3390/e25101464

**Published:** 2023-10-19

**Authors:** Tomasz Starczewski

**Affiliations:** Departments of Mathematics, Czestochowa University of Technology, 42-200 Częstochowa, Poland; tomasz.starczewski@pcz.pl

**Keywords:** decision making, multi-criteria decision, pairwise comparison, inconsistency indices, AHP, Monte Carlo simulation, 90B50, 91B06, 90C29, 62C99

## Abstract

The Analytic Hierarchy Process (AHP) is a widely used used multi-criteria decision-making method (MCDM). This method is based on pairwise comparison, which forms the so-called Pairwise Comparison Matrix (PCM). PCMs usually contain some errors, which can have an influence on the eventual results. In order to avoid incorrect values of priorities, the inconsistency index (ICI) has been introduced in the AHP by Saaty. However, the user of the AHP can encounter many definitions of ICIs, of which values are usually different. Nevertheless, a lot of these indices are based on a similar idea. The values of some pairs of these indices are characterized by high values of a correlation coefficient. In my work, I present some results of Monte Carlo simulation, which allow us to observe the dependencies in AHP. I select some pairs of ICIs and I evaluate values of the Pearson correlation coefficient for them. The results are compared with some scatter plots that show the type of dependencies between selected ICIs. The presented research shows some pairs of indices are closely correlated so that they can be used interchangeably.

## 1. Introduction

The multi-criteria decision problems are characterized by the existence of many conflicting criteria. Forming an accurate assessment of priorities’ weights which are proportional to the utility of available alternatives is, in these situations, difficult for decision makers. Therefore, many techniques, which play a role in multi-criteria support tools, are designed. Among these techniques is the Analytic Hierarchy Process (AHP) [1,2].

The AHP takes advantage of the pairwise comparisons, because, according to scientists’ opinions, a comparison of two alternatives under one criterion is the easiest way of comparing them [3]. The results of comparisons of all alternatives (compared to each other) under one criterion are collected in the pairwise comparison matrix (PCM). Such a matrix is the basis for assessing priorities’ weights. These weights are proportional to the values of a utility function of a decision maker and they are collected in a priority vector (PV) [2].

For obtaining a PV from the PCM, one of the prioritization methods can be used [4,5]. However, the results obtained using various methods are usually not the same and may contain errors when the PCM is not consistent. Therefore, the inconsistency of PCMs is measured, and matrices with a high degree of inconsistency are recommended to be improved. In order to measure the inconsistency of PCMs, the inconsistency index (ICI) has been proposed by the inventor of the AHP [2]. However, some authors have found other measures of the inconsistency of PCMs, which can indicate the size of errors in the PV in a more suitable way.

In the literature, one can encounter many definitions of ICIs [6]. Some of them are connected with a certain prioritization method, while the others are based on the consistency definition. All indices are designed to measure the deviation of the PCM from a consistent matrix. Due to this fact, it seems to be obvious that the values of various indices may be correlated. However, this correlation is not the same for various pairs of indices. Some authors have algebraically shown the relationship between Saaty’s Index and the Geometric Index [7], and some other pairs of indices [8]. On the other hand, the numerical study on the relationship between some indices has been conducted [9,10] and strong dependencies between many ICIs have been shown. In my research, I would like to present the values of the correlation coefficient between selected pairs of indices, mainly involving indices based on the triads idea. However, my results have also been compared with results for traditional indices. The outcomes are obtained in Monte Carlo simulations, which are similar to investigations previously conducted by some authors [9,10,11,12,13]. Some plots that show the relationship between selected pairs of indices are also presented. The method of presenting results is somehow different than this in [9].

## 2. Preliminaries

Let us imagine that a decision maker (DM) has to solve a multi-criteria decision problem, which consists of selecting one or more from available *n* alternatives or ranking these alternatives due to its priorities. Let us assume that the alternatives can be scored due to some criteria. Let us assume existing values of weights $w_{i}$ for i=1,2,…,n, which are proportional to the values of the decision maker’s utility function for the *i* alternative under a certain criterion. Therefore, we assume the existence of a priority vector (PV):(1)$$w=[w_{1},w_{2},\ldots ,w_{n}]$$

An aim of the AHP is to support a decision maker in the estimation of the values of such a PV. To achieve this purpose, the decision maker is asked to compare alternatives in pairs—each alternative with each alternative under each criterion. In such a way, PCMs arise. Therefore, elements of the PCM are treated as ratios of the PV elements:
(2)$$a_{ij}=\frac{w_{i}}{w_{j}}$$ An “ideal” matrix obtained in the pairwise comparison process is an n×n matrix, of which elements satisfy Formula (Equation 2). This matrix is called a matrix of priority ratios (MPR). However, in reality, the elements of PCMs are only approximations of these ratios. The elements of a real PCM are given by the decision maker, so they usually contain some inaccuracies or errors [4,5,14,15]. Therefore, the common assumption in the simulation research is that the elements of the PCM satisfy following formula:
(3)$$a_{ij}=\frac{w_{i}}{w_{j}}\cdot \varepsilon_{ij}$$ In this formula, the ratios of weights are multiplied by a perturbation factor (PF), marked as $\varepsilon_{ij}$. This PF is a random variable, of which the value is usually obtained from probability distributions such as: uniform, gamma, trunc-normal, log-normal, beta, Cauchy, Laplace or Fisher–Snedecor [5,13,14,16]. Despite the justification of using each probability distribution, it turns out that it has no significant influence on the distribution of errors in the obtained PV [17]. Therefore, in our simulation, we use only some assorted distribution.

In the real world, drawbacks of PCMs cannot be observed [5]. However, it can be verified if a matrix given by the decision maker does not satisfy Formula (Equation 2). For this purpose, a notion of the consistency of the PCM was stated [2]:

**Definition** **1**(Consistency of PCM)**.** *The PCM of size n×n is called consistent when its elements $a_{ij}$, $a_{ik}$ and $a_{kj}$ for i,j,k=1,2,…,n satisfy condition:*
(4)$$a_{ij}=a_{ik}\cdot a_{kj}$$

It is easy to show that the following theorem is satisfied [2,14]:

**Theorem** **1.**
*The PCM is consistent if and only if its elements satisfy Equation (Equation 2).*


Therefore, the consistent matrices are treated as error-free matrices. Although it is not always true [11,15], the only used indicators of PCM quality in the AHP praxis are inconsistency indices, which are a “measure” of the deviation from a consistent PCM.

The next property of PCM, which is usually required in the AHP, is reciprocity. This is given by the next definition.

**Definition** **2**(Reciprocal PCM)**.** *The PCM of size n×n is called reciprocal when its elements $a_{ij}$ for i,j=1,2,…n satisfy a condition:*
(5)$$a_{ij}=\frac{1}{a_{ji}}$$

The next theorem is also satisfied:

**Theorem** **2.**
*A consistent matrix is a reciprocal matrix.*


In an AHP praxis, the reciprocity of PCM is enforced by procedure. The decision maker fills the PCM with elements above the main diagonal, and the elements below the main diagonal are calculated in accordance with Formula (Equation 5). This praxis is mainly caused by the impossibility of evaluating the values of Saaty’s inconsistency index for non-reciprocal matrices [18].

## 3. Inconsistency Indices

The ICI, which was originally introduced in the AHP, is called Saaty’s Index (SI). This index is closely related to the prioritization procedure called the right eigenvector method [2]. The SI is given by the following definition.

**Definition** **3**(Saaty’s Inconsistency Index)**.** *Let a given PCM be a reciprocal matrix of size n×n. Let us mark $\lambda_{\max}$ as its maximal eigenvalue. Then, the SI is defined by formula:*
(6)$$SI=\frac{\lambda_{\max}-n}{n-1}$$

The SI is properly defined for a reciprocal matrix. It is equal to 0 if and only if the reciprocal PCM is consistent and it is positive in other cases. However, these two conditions are satisfied by all indices introduced in the literature. Moreover, it is quite obvious that infinitely many functions exist, which satisfies these two conditions [13]. Therefore, in the literature, a lot of definitions of ICIs have appeared.

The second widely used ICI is the Geometric Index (GI). This index has been introduced by Crawford and Williams [14] and developed by Aguaron and Moreno-Jimenez [7]:

**Definition** **4**(Geometric Inconsistency Index)**.** *Let $a_{ij}$ for i,j=1,2,…,n means element of the PCM, and let $v_{i}$, $v_{j}$ mean estimations of the values of the PV given by formula: $v_{i}=\prod_{k=1}^{n}a_{ik}$. Then, the GI is defined by formula:*
(7)$$GI=\frac{2}{\left(n-1)(n-2\right)}\sum_{i<j}\ln^{2}\left(\frac{a_{ij}v_{j}}{v_{i}}\right)$$

The GI is closely related to the prioritization method called the Row Geometric Mean Method (RGMM). It has been shown [7,14] that this index is an unbiased estimator of the variance of the PF for the PV obtained by the RGMM.

The next indices presented here are more related to the definition of consistency than to any prioritization method. Such an index has been introduced by Salo and Hamalainen [19,20]. It is given by the following definition:

**Definition** **5**(Salo–Hamalainen Inconsistency Index)**.** *Let $a_{ij}$ for i,j=1,2,…,n means element of the PCM, and let ${\bar{r}}_{ij}=\max\left\{a_{ik}a_{kj}:k=1,2,\ldots ,n\right\}$ and ${\underline{r}}_{ij}=1/{\bar{r}}_{ij}$. Then, the SHI is defined by formula:*
(8)$$SHI=\frac{2}{n(n-1)}\sum_{i>j}\frac{{\bar{r}}_{ij}-{\underline{r}}_{ij}}{\left(1+{\bar{r}}_{ij}\right)\left(1+{\underline{r}}_{ij}\right)}$$

One can notice that the SHI uses extreme differences between the values of products $a_{ik}a_{kj}$, which arise from the consistency definition. It is worth noticing that ${\underline{r}}_{ij}=\min\left\{a_{ik}a_{kj}:k=1,2,\ldots ,n\right\}$, in fact.

The next index, which is also connected with consistency definition, is the Koczkodaj Index (KI) [21]. The definition of the KI is connected to the triad notion. When I describe triad, I have in mind any trio of elements: $a_{ij},a_{ik},a_{kj}$ of the PCM. Koczkodaj defines triad inconsistency as follows:

**Definition** **6**(Single triad inconsistency according to Koczkodaj)**.** *Let $a_{ij},a_{ik},a_{kj}$, for i,j,k=1,2,…,n is a triple of PCM elements (called triad). Then, the inconsistency of the triad is equal:*
(9)$$TI\left(i,k,j\right)=\min\left\{\left|1-\frac{a_{ij}}{a_{ik}a_{kj}}\right|,\left|1-\frac{a_{ik}a_{kj}}{a_{ij}}\right|\right\}$$

Then, the KI is defined as maximum of inconsistency of all triads of the upper triangular part of the PCM:(10)$$KI=\max_{i<k<j}\left\{TI\left(i,k,j\right)\right\}$$

When the other function of single triad inconsistency combining is applied, the other inconsistency index is obtained. For example, following Grzybowski [11], we can use an average function and define the index called the ATI:(11)ATI=1n3∑i<k<jTI(i,k,j)

Using median [13], this, in turn, leads to the index MTI, expressed by formula:(12)MTI=Me{TI(i,k,j):i<k<j}

One can obtain the next indices based on the triads idea by applying the other single triad inconsistency definition (Equation 9). For example, following [13], in my research, we take into account the indices based on the single triad inconsistency given by (Equation 13) and (Equation 14):(13)$$TIA\left(i,k,j\right)=\left|\frac{a_{ik}a_{kj}-a_{ij}}{a_{ik}a_{kj}+a_{ij}}\right|$$
(14)$$TIS\left(i,k,j\right)={\left(\frac{a_{ik}a_{kj}-a_{ij}}{a_{ik}a_{kj}+a_{ij}}\right)}^{2}$$

We obtain the definition of the indices KIA, ATIA, MTIA, KIS, ATIS and MTIS, respectively, by substituting the TIA(i,k,j) and TIS(i,k,j) in place of TI(i,k,j) in Formulas (Equation 10)–(Equation 12).

The last ICI presentedhere is an index introduced by Kazibudzki [12]. This index, called the ALTI (Average Logarithm Triad Inconsistency), is also based on the idea of triads inconsistency. However, the inconsistency of a single triad is defined now by formula:
(15)$$LTI\left(i,j,k\right)=\left|\ln\left(\frac{a_{ik}a_{kj}}{a_{ij}}\right)\right|$$ The ALTI is defined as an average inconsistency of all triads of the upper triangular part of the PCM:(16)ALTI=1n3∑i<k<jLTI(i,k,j)

In the literature, there exist many more definitions of ICIs, but due to the limited capacity of the article, only the ones mentioned above are examined and presented here. However, one can find some more research on the relationships between indices here [9,10].

## 4. Simulation Framework

The aim of my research presented in this article is to show the relationship between selected ICIs. In order to achieve this purpose, I have conducted a Monte Carlo simulation. The Monte Carlo simulation allows us to prepare and examine a huge number of matrices with various disturbances, which can have an influence on the ICIs’ values. Next, the values of indices have been compared and a value of the correlation coefficient has been calculated for each pair of indices.

The PCMs in my simulation are generated in accordance with Formula (Equation 3). The values of “true” PVs are firstly produced by a random number generator with uniform p.d. (probability distribution) and one randomly chosen element is multiplied by an integer from 2 to 6. This procedure allows us to avoid "flat" vectors. Then, the values of the MPR are calculated according to Formula (Equation 2), and next, each value of the MPR is multiplied by the PF, as in Formula (Equation 3). To obtain more realistic values of the PCM, one large mistake is also inserted. This kind of error, called a big error (BE), is obtained by multiplying one selected element of the matrix by a natural number no greater than 5 (BE=1 mean lack of big error). This kind of error is introduced for simulating outliers in the PCM, which can appear because of an improper matrix filling [22], question misunderstandings [15] or even by the procedure itself [13]. The reciprocity of the PCM is enforced by replacement of all elements of the down triangular by the inverse of elements from the upper triangular of the matrix, as in Formula (Equation 5). Next, the values of obtained matrices are rounded to natural number from 1 to 9 and their inverse. This set of numbers is called Saaty’s scale and is usually used in the AHP procedure [2,9]. Although one can also encounter other propositions of rounding scales in the literature [23,24], I do not examine them in my simulation, in order to keep the proper shape of the article. However, because the impact of using Saaty’s scale on the results could be interesting, I also present, latterly, a sample of results gained without scale use. The ICIs’ values for the PCMs calculated in the described simulation are finally calculated. The simulation steps for the basic framework are listed sequentially below. This framework is similar to the ones that exist in the literature [10,11,12,13,16,17,18,25].

Basic simulation framework (BSF):
Random selection of true PV elements from a uniform p.d.Replacement of one randomly selected element of the PV by multiplication (from 2 to 6) of another element.Calculating MPR values in accordance with Formula (Equation 2).Obtaining a PCM by multiplying the values of the MPR by PFs from a selected p.d. as in (Equation 3).Insertion of the BE to a randomly selected element of the PCM.Calculating elements of the PCM lower triangular in accordance with Formula (Equation 5).Rounding the PCM values to Saaty’s scale.Evaluating values of the ICIs for the obtained PCM.Recording the obtained values in one row of a database.

The PF distribution, by which the MPRs are disturbed in the BSF, is one of the four types of p.d., i.e.,: uniform, gamma, trunc-normal or log-normal (the same number of matrices are disturbed by each p.d.). The parameters of each distribution are matched to gain one of five standard deviation values: σ=0.1,0.2,…,0.5. Moreover, one element of the PCM is multiplied by one of five values of BE=1,2,…,5. As a result, each MPR is disturbed in 100 ways. In our simulation, each parameter configuration is applied for 1000 randomly selected matrices, which results in 100,000 examined matrices (records). The whole simulation is repeated for a number of alternatives (size of matrix): n=3,4,5,6,7,8,9.

The presented BSF is designed in such a way that the PCMs gained by this framework could be treated as PCMs given by a decision maker. The process of generating a “true” PV and "true" PCM (called MPR), and when introducing some perturbations to such a PCM is connected with presumption, means that in the real world, people can assess their priority ratios with a certain accuracy. However, from the purely theoretical point of view, the investigation of correlation between the presented ICIs for totally random matrices could be interesting. In my opinion, such matrices do not occur in the real world, but examining them gives us a deeper view into the issue of correlation between ICIs. Therefore, in the simplified simulation framework (SSF), I generate total random matrices. In this framework, the only assumptions about PCMs (the name PCM is rather undue in this moment) is that the elements of the generated matrices are the numbers from Saaty’s scale, and that the produced matrices are reciprocal. The elements of the upper triangle of the PCM are selected from Saaty’s scale—each with the same probability. Next, the values of the lower triangle are calculated. The steps of the second simulation framework can be specified as below.

Simplified simulation framework (SSF):Random selection of elements of the PCM upper triangle from Saaty’s scale with uniform p.d.Calculating elements of the PCM lower triangle in accordance with Formula (Equation 5).Evaluating values of the ICIs for the obtained PCM.Recording the obtained values in one row of a database.

The loop presented above has been conducted 100,000 times to obtain results which are comparable to the results obtained by the BSF. The simulation has been conducted for the size of PCMs equalling n=3,6,9.

## 5. Results

The results are collected in a database, in which each record contains values of nine ICIs calculated for the same PCM. To obtain values of the correlation coefficient between various pairs of ICIs, the database has been sorted with respect to the values of one index from a selected pair, and next, the values between the first and ninth decile have been split into eight intervals with the same length. In such a way, all results are assigned into ten groups. Next, the average values of both examined indices in the obtained intervals have been calculated. Finally, the Pearson correlation coefficient between obtained means has been calculated, and a scatter plot of the results has been drawn (cf. [11]). To maintain the capacity of the article, most of the plots have been placed in the Appendix B at the end. However, twenty-four plots obtained by BSF (with and without scale) and SSF (with scale) are placed in this chapter. The values of the correlation coefficient connected to these figures are highlighted in appropriate tables.

Firstly, I show the results concerning the correlation between ICIs obtained by the BSF (with Saaty’s scale). These results are collected in Table 1, Table 2, Table 3, Table A1, Table A2, Table A3 and Table A4, which contain values of the correlation coefficient between pairs made from the nine indices described earlier: SI, GI, SHI, KI, ATI, MTI, ATIA, ATIS and ALTI (81 pairs in all). In Table 1, the values of coefficients for indices based on the PCMs of the size 6×6 are collected. Looking at this table, we can see rather high correlation for all pairs of indices, because almost all correlation coefficient values are more than 0.9. However, there are some pairs that show very high correlation close to 1. The very high correlation in Table 1, which is equal 0.999, is seen between the SI and GI. Most of the pairs of indices based on the triads idea also show high correlation. The correlation between all indices, which is defined as the average value of single triad inconsistency (ATI, ATIA, ATIS and ALTI), is not less then 0.985 in Table 1. The SHI also shows very high correlation in pairs with indices based on the triad definition. The values of the correlation of the SHI with indices based on the triad definition, besides the KI, are not less then 0.989 in Table 1.

Table 1, Table 2, Table 3, Table 4, Table 5, Table 6 and Table 7 (as well as Table A1, Table A2, Table A3, Table A4, Table A5 and Table A6 in the Appendix A) also contain average coefficient values for each ICI in the pairs with all remaining indices, as well as a total average value for all examined pairs. The average values for each index are written at the bottom of the columns and on the right of the rows, while the total average is written in the bottom-right corner of the tables. One can notice that the average values at the bottom and on the right, as well as the values for each pair in the upper triangular part and lower triangular part of the tables, are not the same (but they are quite similar). The differences in these results are caused by the procedure, in which the obtained values of the indices are primarily sorted with respect to the values of the first index in the pairs, and then divided on the classes. Due to this fact, the results from columns were sorted with respect to the other index than the values in corresponding rows, which has an effect on the correlation values.

In the presented tables, one can usually observe the lowest values of correlation for the KI. The average value for this index is 0.947 for column and 0.932 for row, and they are the smallest among other average values in Table 1. The highest values of correlation for the KI are observed in pairs with the ATI (0.968–0.979) and the MTI (0.970–0.996). The remaining values of correlation for the KI are less than (by rounding to two decimal places) 0.96. This is especially true in pairs with the SI and GI, where the values are less than 0.89. The SI and the GI also show one of the lowest values of mean correlations. The average values for these two (rounded to two decimal places) are, respectively, equal to 0.96 and 0.97. These two indices tend not to show linear correlation in pairs with indices based on the triad idea. However, the SI paired with the GI shows one of the highest values in Table 1, which equals 0.999. One can notice the same value for the following pairs: SHI and ATIA, SHI and ATIS, ATI and ATIA, and ATIS and ALTI. Whereas, the highest value of correlation is 1 and occurs (apart from pairs of indices with themselves) for the pair SHI → ALTI. One can observe that the values of correlation for various pairs of indices among the SHI, ATIA, ATIS and ALTI are some of the highest values of mean correlation in Table 1, which never decrease to under 0.99. The SHI, ATIA and ALTI also gain the highest mean values of correlation in Table 1.

Similar values can be observed in Table 2, where the values of correlation for the PCMs of size 3×3 are collected. In this table, we also observe the highest value of correlation for the pair ALTI ↔ SHI, but the same value is observed for the pair SI ↔ GMI and for pairs made from the KI, ATI and MTI. This observation is natural, because the indices KI, ATI and MTI are equal for the matrix 3×3 (the definition of these indices converge) and the value of the SI and GI for small errors is proportional [7]. Apart from what was mentioned, the values of the coefficients are usually smaller than the values in the previous table. A total average for Table 2 is equal to 0.945, so it is a bit less than total average in Table 1. The highest values of the mean correlation are connected with pairs made from the SHI, ALTI and ATIA. All pairs of indices based on triad inconsistency have a correlation coefficient no less than 0.9, whereas the indices SI and GI in pairs with the KI, ATI, MTI and ATIA have a correlation of less then 0.9.

Table 3 is prepared for indices obtained for matrices 9×9. In this table, the values of the correlation coefficient are usually higher than in Table 2. The average value for all pairs in this table is equal to 0.97, which is almost the same as that in Table 1. The highest mean values are also connected with the indices ATIA, ALTI, SHI, ATI and ATIS and the lowest value for the KI. The highest value inside Table 3 is obtained for the pair SHI ↔ ALTI, but very high correlation can also be observed for the pairs SHI ↔ ATIS, SHI ↔ ATIA and SHI ↔ ATI. All of these values are greater than or near to 0.99.

The values of the Pearson correlation coefficient for matrices of other sizes obtained by the BSF are collected in the Appendix A in Table A1, Table A2, Table A3 and Table A4. One can notice that the total average of the correlation values in each table is no less then 0.97, so the correlation for other sizes is similarly good, as shown in Table 1, Table 2 and Table 3. The correlation value for the pair SI and GI is close to 1 in each table in the Appendix A. One can observe a very high, usually higher then 0.99, correlation for pairs of indices selected from: SHI, ATIA, ATIS and ALTI.

In this section, the scatter plots prepared for selected pairs of indices are also presented. The scatter plots for indices calculated for matrices 6×6 obtained by the BSF are placed in Figure 1, Figure 2 and Figure 3. Although I present scatter plots for only some pairs of the investigated ICIs in this section, the remaining ones can be found in the Appendix B in Figure A1, Figure A2, Figure A3, Figure A4, Figure A5, Figure A6, Figure A7, Figure A8 and Figure A9. One can also request the author of the article to gain access to the normal size scatter plots presented in the Appendix B. Nevertheless, the scatter plots presented in this section and in the Appendix B show the shapes of the relationship between the investigated inconsistency indices.

In the presented plots, one can observe the monotonic relationships in all cases. The plots (a) in Figure 1 and (a) in Figure 3 show an excellent linear dependence for the pairs GI → SI and SHI → ATIA, respectively (the values of the first indices: GI and SHI are marked on the OX-axis and the values of the second: SI and ATIA-on the OY-axis). The plots in these figures confirm the results from Table 1, because the values of the correlation coefficients between these indices are very high, almost equal 1 (boldfaced values in Table 1). The plot (a) in Figure 2 is not perfectly linear. The values of the SHI increase slightly more quickly at the beginning than at the end of the plots (in comparison to the GI). The correlation coefficient for this relationship is 0.979, so it is slightly smaller than for the previously mentioned pairs. On the other hand, the plots (b) in Figure 1, Figure 2 and Figure 3 show, respectively, the dependencies for the pairs KI → SI, KI → SHI and KI → ATIA, so for the dependencies, which are characterized by a weaker correlation in comparison to those previously mentioned (also boldfaced in Table 1), one can observe no linear relationship in these plots. The increases in the KI at the beginning of the plots do not occur in the same way as the values of the other indices. Such relationships result in relatively low values of the Pearson correlation coefficient for the index KI in pairs with the most presented indices.

As has been described in the previous section, I also conduct simplified simulations, to examine the impact of proposed presumptions on the examined dependencies. One of these simplifications is to neglect the rounding to scale. Although decision makers compare alternatives by nine numbers (language comparisons) in the genuine AHP, there also exist such implementations that the decision maker can compare alternatives by any values from a certain continuous set. Such implementations enable one to make assessments of preferences in more precise way. I have conducted simulations according to the BSF, excluding point number 7. The values of the correlation coefficient obtained in such simulations are collected in Table 4 for matrices 6×6 and in Table 5 for matrices 9×9. In the Appendix A, the analogous Table A5 for matrices 3×3 is placed. The biggest differences between values obtained in the BSF, with the use of scale and without the use of scale, can be observed for the greatest size of examined PCMs-for matrices 9×9. However, the mean values for all examined indices except the KI are not less than 0.9 in Table 5. The biggest differences are observed in pairs with the indices SI, GI and KI. The mean correlation for these three indices decreases by about 3 to 6 percent in comparison with previous tables. The correlation values in Table 5 between remaining indices usually decrease less then 2 percent and it sometimes increase (especially true for MTI). We can do similar observations when we look at Table 4.

Some differences can also be observed when we compare Figure 4, Figure 5 and Figure 6 with Figure 1, Figure 2 and Figure 3. The plots prepared for matrices obtained by the BSF without a rounding procedure show bigger convexity. Similarly as before, the values of correlation for dependencies shown in Figure 4, Figure 5 and Figure 6 are highlighted in Table 4. We can see lower values in most cases in Table 4 in comparison to Table 1.

The next tables are prepared based on the SSF results. The matrices produced in the SSF are entirely random, but they satisfy the reciprocity condition. Such matrices are far from the matrices given by the decision maker, in my opinion. However, the examination of ICIs based on matrices, which spun out of any presumptions, can be interesting from the theoretical point of view. The correlation coefficients for such matrices are collected in Table 6 (for matrices of size 6×6) and Table 7 (for matrices of size 3×3). In the Appendix A is placed the analogous Table A6 (for matrices 9×9). The results for the smallest size of examined matrices (in Table 7) show the biggest differences in comparison to the previously presented tables. The total average value for Table 7 is less then 0.9, so it is the smallest in comparison with all tables presented in this section and in the Appendix A. What is interesting is that the values of correlation in Table 6 are usually a little bigger than the values in Table 1, that is for matrices of size 6×6 obtained on the basis of the BSF. Indeed, a lack of additional presumption causes better correlation between the majority of the examined indices calculated for matrices of large sizes. However, when we look at Table 7, we can observe significantly smaller values of correlation in comparison to Table 2. This effect is also visible when we look at Figure 7, Figure 8 and Figure 9 prepared for matrices 6×6 and Figure 10, Figure 11 and Figure 12 prepared for matrices 3×3 obtained by the SSF. Especially for the latters, the convexity of these plots is bigger than for other plots presented in this section (and the Appendix B). Especially, when we look at plot (b) on Figure 10, which presents the relation for the pair KI → SI, we can see that increasing values of the KI between 0.05 and 0.55 have no impact on the values of the SI. The values of correlation for this relationship, which we can see in Table 7, is 0.669. This is a valid observation in light of our considerations. It occurs that irrespectively of the positive correlation values, the indices have a different significance for the AHP procedure. The errors detected by the KI in interval 0.05–0.55 can not be “visible” for the SI. In turn, when we look at plot (a) in Figure 11, we can notice a small increase in the values of the SHI, while the values of the GI increase between 4 and 6. However, this dependence results in the correlation value 0.926. Therefore, I think that we can say that any ICIs give similar results in a certain AHP problem only at the time when the Pearson correlation values for it are very high, being near 1.

## 6. Conclusions

The presented results show the close relationships between many of the examined indices. These relationships can be observed by the values of the correlation coefficient, as well as by scatter plots. The high correlation coefficient value was observed for the pair of the SI and GI, as well as for some pairs of indices based on the triad definition and connected to the consistency condition (Equation 4). Among the presented indices, some pairs show a practically linear relationship, demonstrating values of the Pearson correlation coefficient higher than 0.99 in all presented cases. Some of the investigated indices also manifest very high correlation with most other indices. Among these indices are SHI, ALTI or ATIA. These observations are important in terms of a practical point of view. Good correlation with most other indices makes it possible to use such indices instead of the others. Therefore, such indices are more universal, in my opinion.

The values of correlation coefficients depend on the assumptions of simulations. The basic simulation produces matrices which are similar to the matrices given by the decision maker. The correlation coefficients for such matrices are usually higher than the values obtained for other examined matrices. This observation gives a deeper insight into the issue considered. It occurs that an impact on the dependencies between various ICIs has assumptions, which are taken on the PCMs. Therefore, even high correlation between ICIs do not always lead to equivalence between them. However, because the monotonicity of dependencies is always preserved, examined indices do not lead to discrepancy.

Although some indices have gained very high values of correlation with most of the investigated indices, we should bear in mind that the presented tables contain values for only nine indices. We can divide the investigated indices into two groups—traditional indices connected with a certain prioritization method and indices connected with the consistency definition. Indices based on a triad idea constitute a majority in the second group. Therefore, it should not be a surprise that some indices in each group are usually nearly correlated. However, we can also observe that some indices are weakly correlated with almost all presented indices, while the others are definitely correlated better.

Our observations can be compared with other observations from the literature. For example, Kazibudzki in [10] has simulated dependencies between four ICIs, among which the SI and GI have been considered. However, the simulations presented in [10] have taken into account the values of total inconsistencies calculated as a mean of inconsistency for all PCMs in the AHP model. He conducted a simulation for various numbers of alternatives and criteria, and he considered the relationships between indices calculated for a singular size of matrices and for combined sizes. In [10], scatter plots contain points corresponding to each gained value of the ICIs, which is in opposition to the plots in this article, which contain average values in classes. Nevertheless, the results presented in [10] also indicate a linear correlation between the SI and GI. It is worth noticing that when we observe all of the results in the plots, some fluctuations appear. However, these fluctuations do not disturb the linear character of the relationships.

The other investigations presented in the literature concern the quality of various ICIs. Here, I have in mind the simulation results presented, for instance in [11,12,13,16,26]. In these papers, the authors deal with the relationship between values of the ICIs and the magnitude of errors in the PV. This issue is very important from a practical point of view, because the ICIs are usually treated as a “measure” of PV errors. For this reason, the results included in the present article should also be compared with those results. Grzybowski and Starczewski, in [13], have shown a high correlation between the magnitude of errors in the PV and the values of indices, which are based on the triads idea, such as ATI, ATIA, ATIS or ALTI. A good correlation can also be observed with SHI, while correlations between errors in the PV and the indices SI, GI or KI are usually weaker. These results are compatible with the results presented here. Indeed, in the present article, we could observe a coincidence of values of indices based on the triad idea, as well as for traditional indices. However, the reader should bear in mind that some indices are better indicators of PV quality, so agreement with values of “better” indices should be perceived as a better feature than agreement with “worse” ones.

The author of the presented article is aware that he does not exhaust the issue of correlation between various ICIs used in the AHP. The investigation presented here concerns only nine indices, and only one correlation coefficient is used. Therefore, the next investigations, which show dependencies between other indices, could be conducted. The other scenarios of disturbing matrices or feature of matrices, such as rounding to various scales, could also be considered.

## Figures and Tables

**Figure 1 entropy-25-01464-f001:**
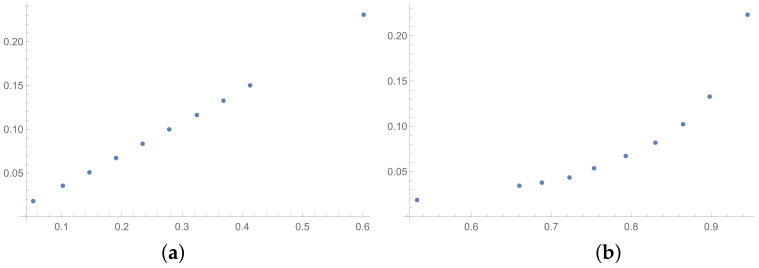
Scatter plots of dependence between: (**a**)-GI (OX-axis) and SI (OY-axis), (**b**)-KI (OX-axis) and SI (OY-axis); values for matrices 6×6 obtained by BSF.

**Figure 2 entropy-25-01464-f002:**
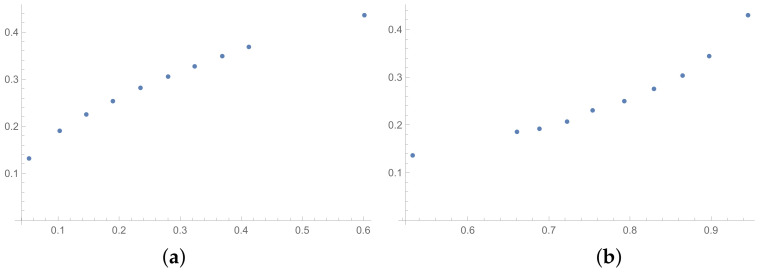
Scatter plots of dependence between: (**a**)-GI (OX-axis) and SHI (OY-axis), (**b**)-KI (OX-axis) and SHI (OY-axis); values for matrices 6×6 obtained by BSF.

**Figure 3 entropy-25-01464-f003:**
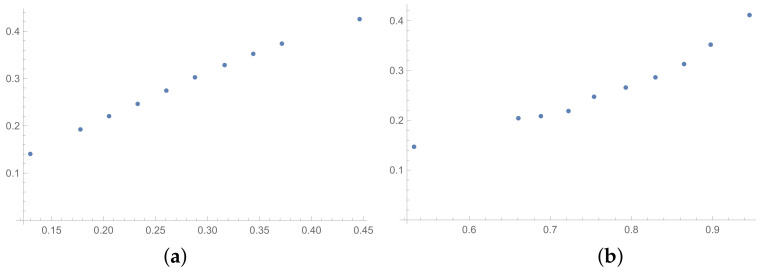
Scatter plots of dependence between: (**a**)-SHI (OX-axis) and ATIA (OY-axis), (**b**)-KI (OX-axis) and ATIA (OY-axis); values for matrices 6×6 obtained by BSF.

**Figure 4 entropy-25-01464-f004:**
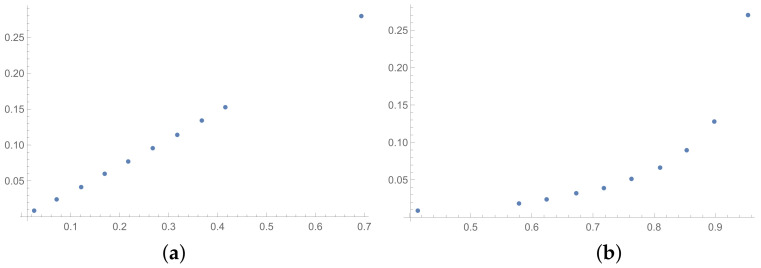
Scatter plots of dependence between: (**a**)-GI (OX-axis) and SI (OY-axis), (**b**)-KI (OX-axis) and SI (OY-axis); values for matrices 6×6 obtained by BSF without rounding to scale.

**Figure 5 entropy-25-01464-f005:**
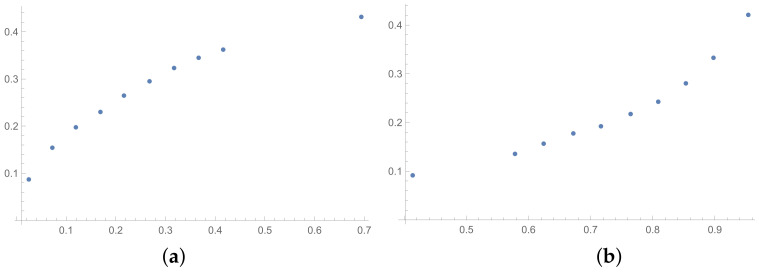
Scatter plots of dependence between: (**a**)-GI (OX-axis) and SHI (OY-axis), (**b**)-KI (OX-axis) and SHI (OY-axis); values for matrices 6×6 obtained by BSF without rounding to scale.

**Figure 6 entropy-25-01464-f006:**
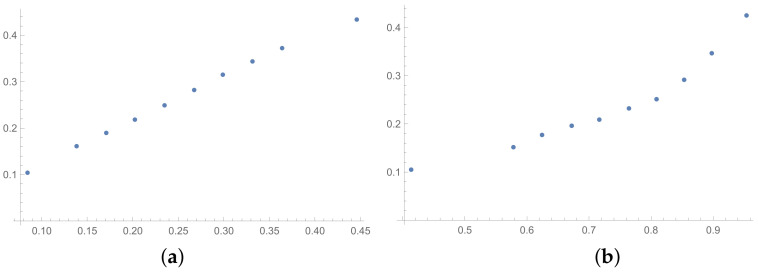
Scatter plots of dependence between: (**a**)-SHI (OX-axis) and ATIA (OY-axis), (**b**)-KI (OX-axis) and ATIA (OY-axis); values for matrices 6×6 obtained by BSF without rounding to scale.

**Figure 7 entropy-25-01464-f007:**
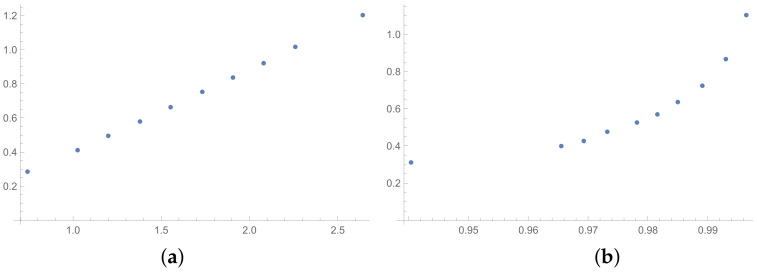
Scatter plots of dependence between: (**a**)-GI (OX-axis) and SI (OY-axis), (**b**)-KI (OX-axis) and SI (OY-axis); values for matrices 6×6 obtained by SSF.

**Figure 8 entropy-25-01464-f008:**
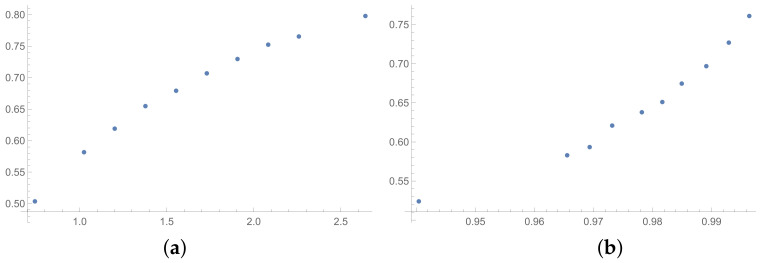
Scatter plots of dependence between: (**a**)-GI (OX-axis) and SHI (OY-axis), (**b**)-KI (OX-axis) and SHI (OY-axis); values for matrices 6×6 obtained by SSF.

**Figure 9 entropy-25-01464-f009:**
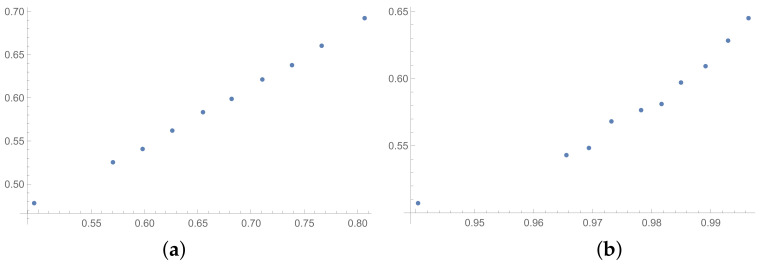
Scatter plots of dependence between: (**a**)-SHI (OX-axis) and ATIA (OY-axis), (**b**)-KI (OX-axis) and ATIA (OY-axis); values for matrices 6×6 obtained by SSF.

**Figure 10 entropy-25-01464-f010:**
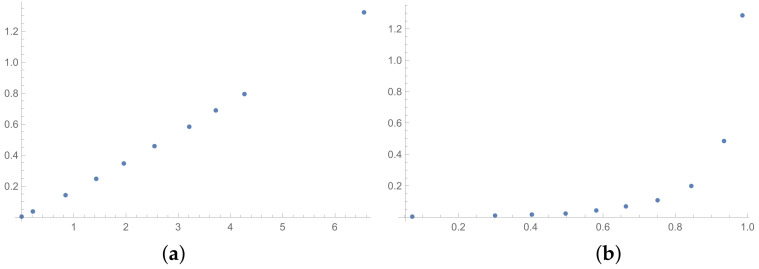
Scatter plots of dependence between: (**a**)-GI (OX-axis) and SI (OY-axis), (**b**)-KI (OX-axis) and SI (OY-axis); values for matrices 3×3 obtained by SSF.

**Figure 11 entropy-25-01464-f011:**
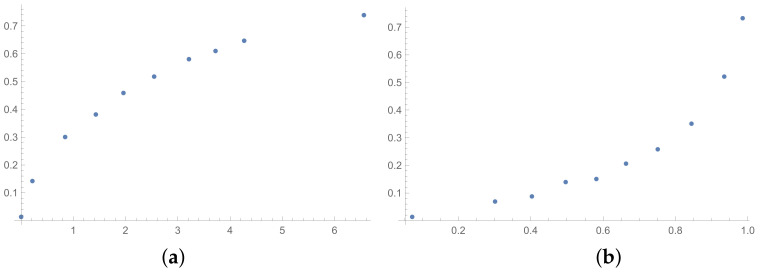
Scatter plots of dependence between: (**a**)-GI (OX-axis) and SHI (OY-axis), (**b**)-KI (OX-axis) and SHI (OY-axis); values for matrices 3×3 obtained by SSF.

**Figure 12 entropy-25-01464-f012:**
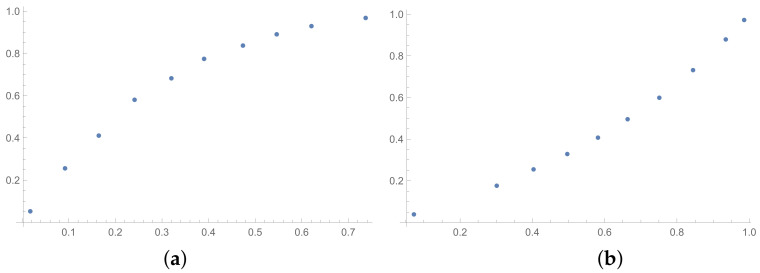
Scatter plots of dependence between: (**a**)-SHI (OX-axis) and ATIA (OY-axis), (**b**)-KI (OX-axis) and ATIA (OY-axis); values for matrices 3×3 obtained by SSF.

**Table 1 entropy-25-01464-t001:** Pearson’s correlation between pairs of ICIs calculated for PCMs of the size 6×6 obtained by BSF.

	SI	GI	SHI	KI	ATI	MTI	ATIA	ATIS	ALTI	Mean
**SI**	1	**0.999**	0.979	**0.864**	0.953	0.917	0.965	0.986	0.977	0.960
**GI**	0.999	1	0.987	0.884	0.961	0.926	0.973	0.991	0.984	0.967
**SHI**	0.970	**0.979**	1	**0.951**	0.997	0.980	0.999	0.996	0.999	0.986
**KI**	0.842	0.859	0.939	1	0.968	0.996	0.947	0.906	0.931	0.932
**ATI**	0.942	0.956	0.992	0.979	1	0.995	0.999	0.989	0.992	0.983
**MTI**	0.946	0.959	0.990	0.970	0.997	1	0.995	0.990	0.988	0.982
**ATIA**	0.957	0.969	**0.997**	**0.970**	0.998	0.987	1	0.995	0.997	0.986
**ATIS**	0.981	0.989	0.999	0.942	0.985	0.955	0.994	1	0.999	0.983
**ALTI**	0.976	0.985	1	0.948	0.993	0.977	0.998	0.999	1	0.986
**Mean**	0.956	0.965	0.988	0.947	0.985	0.972	0.986	0.982	0.985	0.974

**Table 2 entropy-25-01464-t002:** Pearson’s correlation between pairs of ICIs calculated for PCMs of the size 3×3 obtained by BSF.

	SI	GI	SHI	KI	ATI	MTI	ATIA	ATIS	ALTI	Mean
**SI**	1	1	0.965	0.819	0.819	0.819	0.890	0.966	0.952	0.915
**GI**	1	1	0.969	0.827	0.827	0.827	0.897	0.971	0.957	0.920
**SHI**	0.944	0.949	1	0.956	0.956	0.956	0.989	0.992	1	0.971
**KI**	0.808	0.817	0.944	1	1	1	0.987	0.911	0.953	0.936
**ATI**	0.808	0.817	0.944	1	1	1	0.987	0.911	0.953	0.936
**MTI**	0.808	0.817	0.944	1	1	1	0.987	0.911	0.953	0.936
**ATIA**	0.883	0.891	0.981	0.987	0.987	0.987	1	0.966	0.987	0.963
**ATIS**	0.965	0.970	0.996	0.913	0.913	0.913	0.967	1	0.993	0.959
**ALTI**	0.948	0.954	1	0.953	0.953	0.953	0.987	0.992	1	0.971
**Mean**	0.915	0.921	0.978	0.943	0.943	0.943	0.972	0.967	0.979	0.945

**Table 3 entropy-25-01464-t003:** Pearson’s correlation between pairs of ICIs calculated for PCMs of the size 9×9 obtained by BSF.

	SI	GI	SHI	KI	ATI	MTI	ATIA	ATIS	ALTI	Mean
**SI**	1	0.998	0.979	0.859	0.947	0.924	0.961	0.985	0.973	0.958
**GI**	0.998	1	0.988	0.880	0.957	0.934	0.971	0.992	0.983	0.967
**SHI**	0.973	0.984	1	0.937	0.994	0.971	0.998	0.997	1	0.984
**KI**	0.846	0.864	0.935	1	0.981	0.993	0.963	0.907	0.946	0.937
**ATI**	0.937	0.954	0.989	0.959	1	0.992	0.998	0.985	0.992	0.979
**MTI**	0.893	0.913	0.967	0.961	0.988	1	0.979	0.955	0.965	0.958
**ATIA**	0.953	0.968	0.995	0.951	0.998	0.986	1	0.993	0.998	0.982
**ATIS**	0.980	0.990	0.999	0.927	0.983	0.960	0.992	1	0.998	0.981
**ALTI**	0.972	0.983	0.999	0.934	0.993	0.979	0.998	0.999	1	0.984
**Mean**	0.944	0.956	0.983	0.940	0.985	0.974	0.986	0.976	0.984	0.970

**Table 4 entropy-25-01464-t004:** Pearson’s correlation between pairs of ICIs calculated for PCMs of size 6×6 obtained by BSF without rounding to scale.

	SI	GI	SHI	KI	ATI	MTI	ATIA	ATIS	ALTI	Mean
**SI**	1	**0.998**	0.952	**0.790**	0.909	0.920	0.931	0.969	0.954	0.936
**GI**	0.998	1	0.968	0.820	0.926	0.933	0.948	0.982	0.970	0.949
**SHI**	0.927	**0.949**	1	**0.943**	0.995	0.994	0.999	0.991	0.996	0.977
**KI**	0.764	0.795	0.920	1	0.932	0.984	0.917	0.877	0.897	0.898
**ATI**	0.894	0.921	0.994	0.961	1	1	0.997	0.979	0.986	0.970
**MTI**	0.914	0.938	0.995	0.920	0.995	1	0.995	0.985	0.987	0.970
**ATIA**	0.919	0.943	**0.999**	**0.946**	0.997	0.998	1	0.990	0.995	0.976
**ATIS**	0.962	0.979	0.997	0.904	0.972	0.970	0.988	1	0.998	0.974
**ALTI**	0.954	0.972	0.999	0.914	0.987	0.990	0.996	0.999	1	0.979
**Mean**	0.926	0.944	0.980	0.911	0.968	0.976	0.975	0.975	0.976	0.959

**Table 5 entropy-25-01464-t005:** Pearson’s correlation between pairs of ICIs calculated for PCMs of size 9×9 obtained by BSF without rounding to scale.

	SI	GI	SHI	KI	ATI	MTI	ATIA	ATIS	ALTI	Mean
**SI**	1	0.995	0.944	0.761	0.889	0.920	0.915	0.964	0.941	0.925
**GI**	0.994	1	0.966	0.798	0.912	0.937	0.938	0.982	0.964	0.943
**SHI**	0.914	0.948	1	0.921	0.990	0.996	0.998	0.989	0.998	0.973
**KI**	0.737	0.779	0.911	1	0.933	0.987	0.917	0.849	0.900	0.890
**ATI**	0.857	0.901	0.988	0.934	1	0.999	0.996	0.966	0.984	0.958
**MTI**	0.863	0.908	0.989	0.911	0.999	1	0.997	0.973	0.987	0.959
**ATIA**	0.890	0.930	0.996	0.919	0.996	0.999	1	0.984	0.995	0.968
**ATIS**	0.948	0.977	0.995	0.874	0.962	0.974	0.982	1	0.995	0.967
**ALTI**	0.931	0.963	0.999	0.892	0.985	0.992	0.996	0.996	1	0.973
**Mean**	0.904	0.933	0.976	0.890	0.963	0.978	0.971	0.967	0.974	0.951

**Table 6 entropy-25-01464-t006:** Pearson’s correlation between pairs of ICIs calculated for random PCMs of size 6×6 obtained by SSF.

	SI	GI	SHI	KI	ATI	MTI	ATIA	ATIS	ALTI	Mean
**SI**	1	**1**	0.981	**0.857**	0.998	0.967	0.998	0.998	0.997	0.977
**GI**	0.999	1	0.983	0.889	0.998	0.969	0.999	0.999	0.998	0.981
**SHI**	0.966	**0.976**	1	**0.962**	0.994	0.996	0.993	0.992	0.991	0.986
**KI**	0.855	0.865	0.938	1	0.925	0.985	0.916	0.901	0.902	0.921
**ATI**	0.963	0.977	0.999	0.963	1	0.999	1	1	0.995	0.988
**MTI**	0.958	0.971	0.999	0.966	0.990	1	0.991	0.993	0.988	0.984
**ATIA**	0.967	0.981	**0.999**	**0.963**	1	0.996	1	1	0.997	0.989
**ATIS**	0.973	0.985	0.998	0.961	0.999	0.988	1	1	0.999	0.989
**ALTI**	0.991	0.996	0.993	0.927	0.999	0.984	1	1	1	0.988
**Mean**	0.963	0.972	0.988	0.943	0.989	0.987	0.988	0.987	0.985	0.978

**Table 7 entropy-25-01464-t007:** Pearson’s correlation between pairs of ICIs calculated for random PCMs of size 3×3 obtained by SSF.

	SI	GI	SHI	KI	ATI	MTI	ATIA	ATIS	ALTI	Mean
**SI**	1	**1**	0.910	**0.678**	0.678	0.678	0.759	0.844	0.941	0.832
**GI**	1	1	0.931	0.701	0.701	0.701	0.785	0.870	0.957	0.849
**SHI**	0.906	**0.926**	1	**0.889**	0.889	0.889	0.950	0.988	1	0.937
**KI**	0.669	0.699	0.897	1	1	1	0.982	0.901	0.865	0.890
**ATI**	0.669	0.699	0.897	1	1	1	0.982	0.901	0.865	0.890
**MTI**	0.669	0.699	0.897	1	1	1	0.982	0.901	0.865	0.890
**ATIA**	0.748	0.778	**0.954**	**0.982**	0.982	0.982	1	0.967	0.930	0.925
**ATIS**	0.840	0.867	0.990	0.902	0.902	0.902	0.967	1	0.975	0.927
**ALTI**	0.937	0.954	1	0.865	0.865	0.865	0.930	0.973	1	0.932
**Mean**	0.826	0.847	0.941	0.891	0.891	0.891	0.926	0.927	0.933	0.897

## Data Availability

Not research data used.

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
