# Peer review of "Simulation Research on the Relationship between Selected Inconsistency Indices Used in AHP"

_entropy, 2023, doi:10.3390/e25101464_

Round 1

Reviewer 1 Report

Although the Monte Carlo simulation approach is good, the methodology does not seem to be very appropriate, because PCMs are a means of estimating PV, and not the other way around. The decision maker's input in a decision process with AHP is aij, and the degree of consistency of the PCM depends on the aij. Therefore, it would be more appropriate, simpler and more realistic for the calculation of the inconsistency of a PCM uniformly sampling the aij varying from -9 to 9. Starting from some hypothetical PVs and simulating PFs does not contribute to the process and complicates it unnecessarily.

By the way, it is not necessary that the values of aij be integers, as the authors state in the paper. Examples in the literature show software in which the decision maker establishes the aij values on a continuous scale using sliders or knobs. This adds precision to the decision maker's judgments.

Here are some specific comments:

1. In Section 1, Paragraph 3, it is convenient that the author explain what the inconsistency of the PCM is and why it is important to address the problem of inconsistency.

2. In Section 1, Paragraph 3, the author mentions the "triads idea" without explaining what that concept means.

3. In Section 4, Paragraph 2, it is not clear the need of introducing the big error (BE).

4. Table 1 needs an explanation of what the boldfaced numbers mean.

English must be substantially improved. There are paragraphs that are difficult to understand. In general, English is very poor.

Author Response

Thank you for your factual revison. 

At first, I am sorry for my English. I admit that my English is not good, so my article have been sent to extensive language editing. I hope the manuscript is more understandable now.

Althought I understend that my simulation method may seem to be a little strange it is not use wihout any reason. The manner of generating PCMs, which I have used in my simulation, allows me to gain similar values to this one used in AHP in my opinion. It is rather obvious that the matrices given by decision maker are never random. Althought we can not expect the values of PCM will be exactly values of priority ratios, I think we agree that these values fluctuate around precisce values. Therefore, many authors use the formula, in which PF is applied. I also use this formula in my simulation because I hope that it allows me to gain dependencies occuring in reality. However, I admit that it is interesting from theoretical point of view to investigate dependencies, which occure between inconsisteny indices, for random matrices. Therefore, I have performed suitable simulations, in which I generate reciprocal random matrices with values from Saaty's scale. The results are interesting but they do not deviate from previously obtained results. I have completed the article with this results and I also add results obtained during simulation without using rounding to scale.

Furthermore, I compete remaining shortcomings:
- I more clearly explaine what is triad
- I justify introducing "big error"
- I add an explenation for boldfaced number in tables

Please, look at new version of my manuscript and give an opinion about it.

Reviewer 2 Report

See attached file.

English should be improved. See attached file.

Author Response

Thank you for your factual revison. 

At first, I am sorry for my English. I admit that my English is not good, so my article have been sent to extensive language editing. I hope the manuscript is more understandable now.

In accordence with your suggestion, I have made simulation without using scale. Although the results are not exactly the same, the conclusions are similar. Indeed, the correlation values are a bit smaller for indices based on matrices without scale. It is interesting observation and it give dipper insight in dependencies between inconsistency indices. 

I also complete describing of the graphs. I admit that they were poorly described. I made some more investigation and in my new version of manuscript are additional table and graphs. Some of them concern investigation without scale but there are also results for entierly random matrices (the suggestion from one of reviewers). However, I try to describe it clearly.

For refering to doubts about shape of the graphs I would like underline that I have calculated values of Pearson corelation coefficient. The obtained values are usually rather high but only in some cases, it was near to 1. Therefore, the exactly linear shape of the graphs is not obtain in each plot. I think, it is even more visible on new graphs. The worst values of correlation coefficient presented in new tables is 0.669 and for this value the shape of the graph is far from linear. The shape of graphs for correlation less than 0.9, which was presented earlier, is obvious not exacltly linear.

I am not sure if I have enough explained for your questions. If you have any doubts, you can query again. Please, familiarize yourself with new version of my manuscript and give your opinion about it.

Round 2

Reviewer 1 Report

Of course the values obtained in this second way do not deviate from previously obtained results, since your original Monte Carlo simulation was large enough. It is an interesting result to verify that the values of both simulations coincide. But I consider that it is much more natural to simulate the real process of assigning comparisons by the decision maker, and therefore start with the paired comparisons as input to the simulation, and then compare the results of this simulation with the theoretical modeling of such comparisons. from real preference scales.

I think the paper is noticeably clearer after the recent modifications, and as far as I'm concerned it can perfectly be published in its current state.

Reviewer 2 Report

The author has dealt with all my suggestion and hence I reccomend to accept the paper for publication. 

Just two small typos:

- line247: A reference is misled here.

- lines 403, 404: "he conducts", "he considers".